# Vaccine Hesitancy towards the COVID-19 Vaccine in a Random National Sample of Belgian Nursing Home Staff Members

**DOI:** 10.3390/vaccines10040598

**Published:** 2022-04-12

**Authors:** Marina Digregorio, Pauline Van Ngoc, Simon Delogne, Eline Meyers, Ellen Deschepper, Els Duysburgh, Liselore De Rop, Tine De Burghgraeve, Anja Coen, Nele De Clercq, An De Sutter, Jan Y. Verbakel, Piet Cools, Stefan Heytens, Laëtitia Buret, Beatrice Scholtes

**Affiliations:** 1Research Unit of Primary Care and Health, Department of General Medicine, Faculty of Medicine, University of Liège, 4000 Liège, Belgium; simon.delogne@student.uliege.be (S.D.); laetitia.buret@uliege.be (L.B.); beatrice.scholtes@uliege.be (B.S.); 2Department of Diagnostic Sciences, Faculty of Medicine and Health Sciences, Ghent University, 9000 Ghent, Belgium; eline.meyers@ugent.be (E.M.); piet.cools@ugent.be (P.C.); 3Biostatistics Unit, Faculty of Medicine and Health Sciences, Ghent University, 9000 Ghent, Belgium; ellen.deschepper@ugent.be; 4Department of Epidemiology and Public Health, Sciensano, 1050 Brussels, Belgium; elza.duysburgh@sciensano.be; 5EPI-Centre, Department of Public Health and Primary Care, KU Leuven, 3000 Leuven, Belgium; liselore.derop@kuleuven.be (L.D.R.); tine.deburghgraeve@kuleuven.be (T.D.B.); jan.verbakel@kuleuven.be (J.Y.V.); 6Department of Public Health and Primary Care, Faculty of Medicine and Health Sciences, Ghent University, 9000 Ghent, Belgium; anja.coen@ugent.be (A.C.); nele.declercq@ugent.be (N.D.C.); an.desutter@ugent.be (A.D.S.); stefan.heytens@ugent.be (S.H.); 7Nuffield Department of Primary Care Health Sciences, University of Oxford, Oxford OX2 6GG, UK

**Keywords:** COVID-19, vaccine hesitancy, nursing home, staff, Belgium, COVID-19 vaccination

## Abstract

In Belgium, nursing home staff (NHS) and residents were prioritised for COVID-19 vaccination. However, vaccine hesitancy may have impacted vaccination rates. In this study, a random stratified sample of NHS (*N* = 1142), vaccinated and unvaccinated, completed an online questionnaire on COVID-19 vaccine hesitancy (between 31 July and 15 November 2021). NHS who hesitated or refused the vaccine were asked for the main reason for their hesitation/refusal. Those who hesitated, but eventually accepted vaccination, were asked why they changed their minds. Overall, 29.5% of all respondents hesitated before accepting vaccination, were still hesitating, or refused vaccination. Principal reasons were fear of unknown future effects (55.1% of vaccinated participants that hesitated and 19.5% who refused), fear of side-effects (12.7% of vaccinated participants that hesitated and 12.2% who refused), and mistrust in vaccination (10.5% of vaccinated participants that hesitated and 12.2% who refused). For vaccinated participants who hesitated initially, protecting the vulnerable was the main reason they changed their minds. Given this degree of fear and proposals to mandate vaccination among healthcare workers, communicating with NHS on the safety and efficacy of the vaccine should be prioritised.

## 1. Introduction

Mortality due to COVID-19 increases with age [1]. In Belgium, nursing home residents (NHR) were particularly affected by COVID-19, and this population accounted for the majority of fatalities [2].

Since the beginning of the pandemic, efforts have been made to rapidly produce an effective and safe vaccine. Due to the vulnerability of the elderly to COVID-19, Belgian authorities prioritised nursing home staff members (NHS) and NHR. The first Belgian vaccination campaign in nursing homes took place between 28 December 2020 and 24 March 2021 [3]. Data from the first analysis of a national study on the seroprevalence of SARS-CoV-2 antibodies showed that, during the COVID-19 vaccination campaign (February–March 2021), NHS in Belgium had a lower vaccination rate for COVID-19 compared to NHR, with 48% and 68% fully vaccinated with BNT162b2 (Pfizer-BioNtech), respectively [4]. This disparity in vaccination rates between NHS and NHR has previously been observed for other vaccines, such as the influenza vaccination [5].

A vaccine is only truly effective if it is administered to the majority of the population and particularly to those most at risk. This is known as herd immunity [6,7]. Vaccine hesitancy was described by the World Health Organization (WHO) as one of the ten major threats to global health in 2019 and has been defined as the “delay in acceptance or refusal of safe vaccines despite availability of vaccination services” [8]. A better understanding of the proportion of NHS that hesitate or refuse the vaccine, the reasons for this, and what contributes to an eventual acceptance of the vaccine would allow nursing home managers, politicians, and other stakeholders to identify and prioritise strategies to increase vaccine acceptance among NHS. Previous studies have investigated COVID-19 vaccine hesitancy in diverse populations, such as hospital staff, healthcare workers, or the general population in various countries [9,10,11,12,13,14,15,16,17,18,19]. However, to the best of our knowledge, no study has explored COVID-19 vaccine hesitancy among NHS in Belgium.

The present study aimed to identify the proportion of NHS in Belgian nursing homes (NH) that hesitated or refused to get vaccinated against COVID-19, the principal reasons for their hesitation/refusal, and the principal reasons they eventually decided to get vaccinated (for participants that hesitated but eventually accepted vaccination).

## 2. Methods

### 2.1. Study Design and Population

The present study is a sub-study of a national study (SCOPE) in which the prevalence of SARS-CoV-2 antibodies was assessed among Belgian NHS and NHR using a rapid antibody test. The study design, including sample size calculation, has been described in the study protocol [20]. Briefly, in this prospective cohort study, 69 NH, geographically and demographically representative for NH in Belgium, were randomly selected and a total of 1656 NHR and 1380 NHS were randomly recruited to achieve a random sample of 20 NHS and 24 NHR per NH. Six testing visits (V) were carried out in February, April, June, August, October, and December 2021. Each testing visit involved performing a rapid anti-body test and participants were asked to complete an online questionnaire. The present paper reports on a nested study that included only the NHS participating in the SCOPE study. All NHS participating in the SCOPE study were invited to answer questions about vaccine hesitancy that had been added to the usual questionnaire in August (or in October 2021 if the participant had been absent at the August testing visit).

### 2.2. Ethics Approval and Consent to Participate

The SCOPE study was approved by Ghent University Hospital Ethics Committee (reference number BC-08719) on 11 December 2020. An amendment for the nested study was approved on 13 July 2021 (reference number BC-08719-AM02). The study was conducted according to the approved protocol and the principles outlined in the Declaration of Helsinki. At the start of the study (February 2021), each participant was informed of the goal of the study, the intended use of the collected data and the pseudonymization of their data, all participants signed an informed consent form. Participants did not receive any gift or financial reward for the time invested.

### 2.3. Data Collection

In all six testing phases, an online questionnaire (LimeSurvey version 3.22, Hamburg, Germany) was completed by each participant (NHS) within one week after the antibody testing for the SCOPE study.

A baseline questionnaire was completed during the first visit (V1; between 1 February and 24 March 2021). In this baseline questionnaire, participants were questioned about various individual characteristics (age, sex, type of job), presence of one of the following comorbidities (cardiovascular disease, diabetes, hypertension, immunosuppression, severe renal/lung/cardiac disease, active cancer), their influenza vaccination status in 2020 (yes/no), and COVID-19 vaccine (yes/no, type of vaccine, date of vaccination). The latter information was collected at each visit (from V1 to V6).

During the fourth phase of testing (V4; between 27 July and 29 September 2021), single-choice questions about vaccine hesitancy were added to the usual follow up questionnaire. Participants were asked if they were vaccinated and if so, whether they had hesitated before getting vaccinated. For participants that were not vaccinated, the main reason for refusal was asked. The main reason for hesitation/refusal and the principal reason participants changed their minds were explored using single-choice questions with thirteen and fourteen different answer options respectively. Each question was mandatory but there was an option at each stage to indicate if they preferred not to answer. The questionnaire was inspired by the items described by Larson and colleagues [21]. Several of the authors of this paper (with diverse professional profiles) were directly involved in the elaboration of the questionnaire.

Figure 1 shows how the questions were organised. Due to absences in August, questions were also asked during the fifth visit (V5; between 29 September and 15 November 2021) for those participants that had not completed the questionnaire during V4. Four profiles for vaccine hesitancy were coded as followed: vaccinated, no hesitation before; vaccinated, hesitation before; not yet vaccinated or soon, (still) hesitating; refused vaccination.

### 2.4. Data and Statistical Analysis

The chi-squared test (χ2) was used to test the independence between individual characteristics and the profile of hesitancy. The degree of this association was evaluated with Cramer’s phi (*Vc*).

The odds ratios were estimated based on a GEE analysis with exchangeable covariance structure, taking the clustered nature of the staff within NHs into account. For this purpose, a new binary variable was created for vaccine hesitancy with non-hesitant participants (vaccinated NHS that did not hesitate before vaccination) on one side and hesitant participants (vaccinated NHS that hesitated, unvaccinated NHS still hesitating or unvaccinated NHS refusing vaccine) on the other. Collinearity between parameters was assessed using a variation inflation factor (VIF). Odds ratio (OR), adjusted OR for all covariates (multivariate analysis), and 95% confidence interval (95% CI) are reported.

In the present study, hesitation was analysed separately from refusal, when possible. Participants that were uncertain about getting vaccinated were asked the main reason for this hesitancy. For this analysis, when participants chose ‘other’, the reason given was manually analysed with attribution to a proposed item if possible. When the same reason was given frequently, a new item was created, and this was the case for the item “pregnancy/future pregnancy” in our analysis.

Finally, participants who hesitated initially but eventually got vaccinated were asked the main reason they changed their minds. The section ‘other’ was analysed as described above and new items were generated: “perceived government pressure” and “supplementary information received from the nursing home”.

All *p*-values were two-sided and a *p*-value of ≤0.05 was considered statistically significant. Statistical analyses and graphical representations were performed by using GraphPad Prism software (version 9 for Windows, GraphPad Software, San Diego, CA, USA), R (version 4.1.1), Microsoft Excel 2019, and MatLab (version 2020b).

## 3. Results

### 3.1. Description of the Study Cohort

Of the total cohort (*N* = 1368), 1142 (83.5%) NHS completed the questionnaire on vaccine hesitancy (1076 during V4 and 66 during V5). The majority of respondents were over 40 years old (*N* = 629; 55.1%), female (*N* = 961, 84.2%), Flemish (*N* = 678; 59.4%), and Walloon (*N* = 384; 33.6%) and worked as nurses or care workers (*N* = 582; 51%). The majority, *N* = 893 (78.2%) reported not having any co-morbidities. Concerning status of influenza vaccination in 2020, the distribution of the NHS was split almost evenly between those who reported being vaccinated and those who reported not being vaccinated (Table 1).

### 3.2. Distribution of Nursing Home Staff Members by Vaccine Hesitancy Question (Profile) and Association with Individual Characteristics

Overall, the majority (*N* = 1081; 94.7%) of participants reported being vaccinated against COVID-19. Among the hesitant participants, we identified three profiles: vaccinated participants who hesitated before vaccination, non-vaccinated participants who were still hesitating, and NHS that refused to get vaccinated.

Concerning vaccine hesitancy, 25.5% of vaccinated NHS hesitated before vaccination and 29.5% of the total cohort that hesitated were still hesitating or refused vaccination. Among the 5.3% of unvaccinated NHS, the majority (67.1%) refused to get vaccinated, 27.9% were still hesitating, and 5% were going to be vaccinated soon (Table 1).

There were differences in individual characteristics in terms of vaccine hesitancy profiles (Table 2). For this analysis, NHS who will be vaccinated soon (*N* = 3) were grouped with NHS still hesitating to get vaccinated (*N* = 17). Age (*p* < 0.0001), gender (*p* = 0.006), region (*p* < 0.0001), type of job (*p* = 0.0008), and previous influenza vaccination (*p* < 0.0001) were all associated with vaccine hesitancy profile. Furthermore, the degree of association was the highest for previous influenza vaccination (*Vc* = 0.33).

To analyse which groups would be more or less likely to hesitate, we compared participants who did not hesitate before vaccination with those who did hesitate, were still hesitating, or refused vaccination (Table 3). Older NHS, male, and NHS working in non-medical fields had a smaller odds of vaccine hesitancy than female and NHS working in medical fields, respectively (OR 0.96, 95% confidence interval (CI): 0.94,0.97; OR 0.46, 95% CI: 0.29,0.75; OR 0.68, 95% CI: 0.49,0.95). Moreover, NHS working in Wallonia had a higher odds of vaccine hesitancy compared to their Flemish counterparts, being 2.22 times more likely to hesitate to get vaccinated (OR Wallonia 2.22, 95% CI: 1.57, 3.12). Finally, NHS that did not report an influenza vaccination in 2020 had a higher odds of vaccine hesitancy than those who reported themselves to be vaccinated (OR 3.86, 95% CI 2.81, 5.3).

### 3.3. Principal Reasons for Hesitation/Refusal of COVID–19 Vaccination and Comparison between the Different Profiles Identified

Those participants that were uncertain about getting vaccinated were asked the main reason for their hesitancy (Figure 2). For vaccinated NHS who hesitated before getting vaccinated, the main reason was fear of unknown future effects with 55.1% (*N* = 152) of answers. For NHS who will be vaccinated soon or were still hesitating, the principal reason was the fear of side-effects, with 66.7% (*N* = 2) and 47.1% (*N* = 8) of answers, respectively. Finally, 31.7% of NHS who refused to get vaccinated preferred to not answer the question (*N* = 13).

The three principal reasons for hesitation/refusal among vaccinated NHS who hesitated and NHS who refused to get vaccinated were very similar. The only difference was that the most common answer for NHS who refused was that they preferred not to answer (*N* = 13, 31.7%). The three main reasons were: fear of unknown future effects (55.1% (*N* = 152) of vaccinated NHS that hesitated and 19.5% (*N* = 8) NHS who refused), fear of side-effects (12.7% (*N* = 35) of vaccinated NHS that hesitated and 12.2% (*N* = 5) NHS who refused), and mistrust in vaccination (10.5% (*N* = 29) of vaccinated NHS that hesitated and 12.2% (*N* = 5) NHS who refused).

### 3.4. Principal Reasons Given by Participants, That Hesitated but Eventually Accepted Vaccination, to Decide to Get Vaccinated

Finally, participants who hesitated at first but eventually got vaccinated were asked the main reason they changed their minds (Figure 3). The three main reasons NHS changed their minds were that: they felt the need to be vaccinated to protect vulnerable people (27.2% (*N* = 75) of answers); a friend/family member/health professional had recommended it to them (13% (*N* = 36) of answers); and they had needed more time to get informed (12.7% (*N* = 35) of answers). When considering the relationship between the main reason why NHS hesitated to get vaccinated and the principal reason they changed their minds (Figure 4), protection of vulnerable people was still the main reason NHS changed their minds regardless of the reason for hesitation.

## 4. Discussion

In this study, we showed that 29.5% of the total cohort of NHS hesitated, are still hesitating, or refused the COVID-19 vaccination. While it is difficult to compare results across countries/populations, together with the few studies that exist on the subject, our results are coherent with those of other studies [9,10,11,17,22]. Despite a high vaccination rate of Belgian NHS, among the unvaccinated participants, the majority stated that they refused to get vaccinated (*N* = 41, 67.1%) compared to NHS who were still hesitating (*N* = 17, 27.9%) or who have hesitated and will be vaccinated soon (*N* = 3, 5%). The main reasons for refusal were similar to those for hesitation, the three principal ones being: the fear of unknown future effects, the fear of side-effects, and mistrust in vaccination.

Similar to our study, a study conducted in the United States prior to the vaccination campaign found that NHS over 60 years of age, or staff who do not work in clinical care, were more willing to receive the vaccine compared to younger groups and staff who work in the clinical setting [22]. The same observations were made among Belgian hospital staff members [17] and the general Belgian population before the vaccination campaign [18,19]. Data presented in other Belgian studies concerning profiles that are particularly vaccine hesitant may also be applicable to our population, such as educational attainment, financial status, or family composition. These profiles have been shown to be associated with status of hesitation regarding COVID-19 vaccination [18]. Other European studies show that factors, such as young age [9,10,11], female gender [9], lack of trust in vaccines [10,11], and fear of short- and long-term side effects [11], are related to COVID-19 vaccine hesitancy. Despite the complexity of comparing the reasons for vaccine hesitancy across countries and time periods, the reasons highlighted in this paper are consistent with other studies [9,10,11,12,13,14,15,17].

Differences observed in our study between the different Belgian regions were also observed among healthcare professionals. On 31 October 2021, full vaccination coverage varied from 72.9% for healthcare workers residing in Brussels and 83.3% in Wallonia to 94.7% for healthcare professionals residing in Flanders [23]. This trend was also shown to be present at the beginning of the vaccination campaign among nursing home staff and a few months later in the general population [23]. In Belgium, the implementation of vaccination campaigns is devolved to the regions. The Belgian governments, at federal and regional levels, have a central role to identify and prioritise strategies to increase vaccine acceptance among NHS. Efforts should be made to reduce regional inequalities in vaccine uptake and cross-regional learning could improve uptake in Wallonia and Brussels.

A study in the general Belgian population previously showed that vaccine hesitancy is higher for COVID-19 vaccines than for other vaccines [19]. For a population that is particularly in contact with vulnerable people, public health messaging (ideally in collaboration with trusted figures, such as nurses or doctors) should communicate the efficacy and safety of the COVID-19 vaccine, as already suggested by others [12,24]. Moreover, subgroups we have identified as being particularly hesitant should be consulted and integrated into vaccination campaign policies, e.g., by providing evidence of the benefit of vaccination in the particular context of NH. Communication by the government and public trust in the government are mediators in combatting misinformation, especially about the safety and efficacy of vaccination among workers in contact with vulnerable people. Indeed, several studies have already shown the increase in vaccine hesitancy due to misinformation about safety [15,16,24]. There is a need to promote information sources such as scientific journals containing reliable evidence-based information, which play a predominant role in health professionals’ attitudes towards the vaccine [15].

As a perspective, qualitative studies that explore in more depth the impact of approaches, information sources, and communication strategies, particularly in this population, would be interesting. Furthermore, exploring the motivations of those who have been vaccinated could give us an insight into the other side of the issue. As it is likely that further booster doses will be necessary, analysing the reasons for hesitancy for these additional doses would allow a better understanding of the concerns of NHS who have already agreed to receive the first two doses.

This national study is the first in Belgium to evaluate vaccine hesitancy among a large cohort of NHS, the reason for their hesitation, and the reason they changed their minds. The NHS cohort was randomly recruited among geographically and demographically representative Belgian NH. Although a social desirability bias could have been introduced due to the nature of this study, funded by the scientific institute of public health (SCIENSANO), participants were already familiar with the study team, the study design, and the questionnaires, when they answered questions about vaccine hesitancy during the fourth and the fifth visits for the wider national study. Indeed, a large participation rate was achieved with 83.5% of the full cohort at baseline.

Nevertheless, there are some weaknesses to our study. First, the list of possible answers for the main reason for hesitation and the reason NHS changed their minds were systematically proposed in the same order, which could have introduced cognitive bias where the participant chooses the first response. Second, we were unable to pilot the short questionnaire before data collection. However, the participation of a multi-disciplinary group in the elaboration of the questionnaire allowed us to assess and improve its validity before its launch [25]. Third, hesitation concerning the type of vaccine was not analysed in this study, as the majority of this population was offered the vaccine BNT162b2 (Pfizer-BioNtech) [4]. Fourth, vaccine hesitancy questions were collected after the vaccination campaign, which restricts our analysis to a very specific context and could have introduced a recall bias. However, our study also shows that NHS who were vaccinated at the time of the questionnaire but had hesitated (25.5% of vaccinated NHS), mostly got vaccinated to protect the most vulnerable people.

Mandatory vaccination among health care professionals has been proposed in Belgium and is under debate. In view of possible policy changes and subsequent public discussion of the topic, vaccine hesitancy is likely to vary over time. In Flanders, vaccine willingness was associated with time of the year, with the highest willingness in July, 2020 compared to August–December, 2020, but those results are also somewhat hypothetical since vaccines were not available in July 2020 [18]. Our data are therefore specific to the time-period during which the data were collected. Indeed, the vaccine passport has been implemented in Belgium since November 2021 and requires proof of either full vaccination, a recent negative test result or COVID-19 recovery to enter restaurants, bars, fitness clubs, etc. [26,27]. Studies have shown that the implementation of a vaccine passport could have negative effects on the motivation to be vaccinated [28,29]. Our data could therefore be influenced by this particular period, i.e., six months into the vaccination campaign and during the implementation of the vaccine passport. These policy proposals were frequently discussed by the media at that time.

Currently, mandatory vaccination among health care professionals has been proposed in Belgium and is under debate. NHS were one of the populations most affected by the COVID-19 crisis and among the first populations to be vaccinated in Belgium. Since then, vaccination strategies and campaigns have continued, but this particular population has not been reassured concerning their initial fears. As an international study has suggested, vaccination campaigns should target specific hesitant audiences and address their concerns [30]. Establishing a clear and open communication strategy by reassuring and communicating about the safety and efficacy of the vaccine should be a priority for governments.

## 5. Conclusions

This study showed that, although a large majority of NHS are vaccinated, 25.5% of this population hesitated before accepting vaccination. Our study supports population-specific communication. In addition to the profiles already determined as associated with vaccine hesitancy (the female gender, being younger), our study showed that NHS that hesitated before being vaccinated and finally changed their minds accepted vaccination mainly to protect the most vulnerable people. In the context of emerging variants and the requirement for booster doses, a communication strategy, adapted to this population in contact with the elderly, is a vital tool to manage this crisis.

## Figures and Tables

**Figure 1 vaccines-10-00598-f001:**
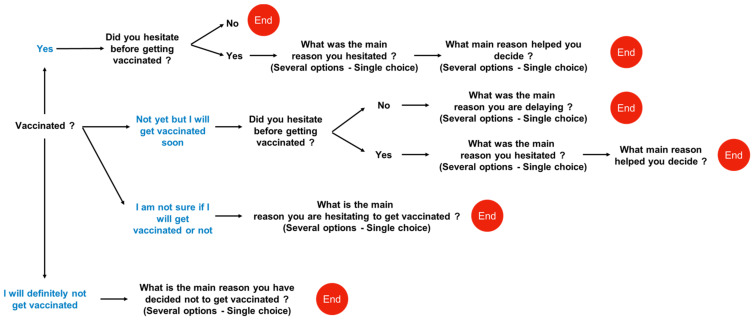
Schematic representation of questions asked to nursing home staff members about vaccine hesitancy in August and October 2021. Questions are schematically organised taking into account the answer to the previous question. Depending on the answer, the next question was asked, or the questionnaire ended.

**Figure 2 vaccines-10-00598-f002:**
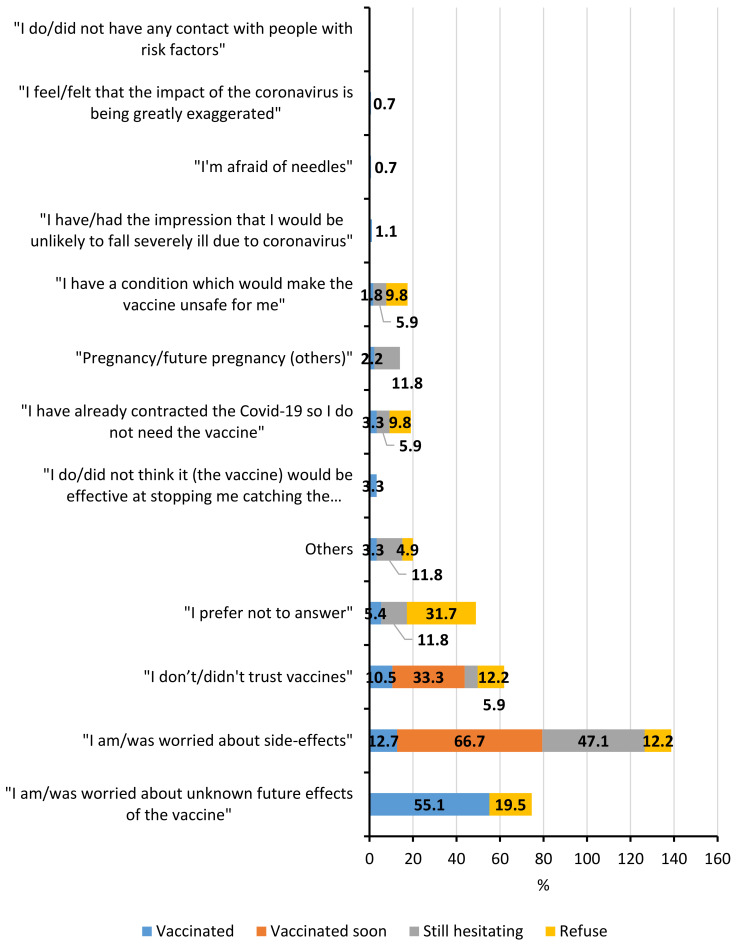
**Main reason for hesitation among nursing home staff members (NHS); distribution by vaccine hesitancy profiles.** Profiles are: vaccinated NHS that hesitated before getting vaccinated (blue, *N* = 276), unvaccinated NHS that plan to get vaccinated soon (orange, *N* = 3), unvaccinated NHS still hesitating (grey, *N* = 17) and unvaccinated NHS that refuse to get vaccinated (yellow, *N* = 41). Data are shown as a percentage of answer in each profile.

**Figure 3 vaccines-10-00598-f003:**
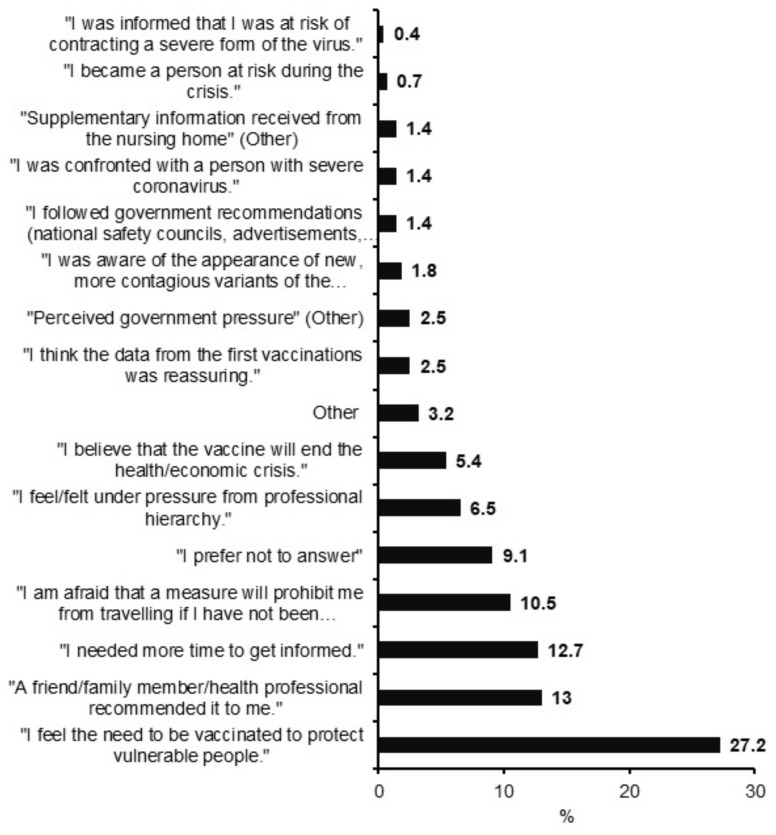
**Main reason to decide to get vaccinated for participants that hesitated and finally got vaccinated** (*N* = 276). Data are shown as a percentage of answer.

**Figure 4 vaccines-10-00598-f004:**
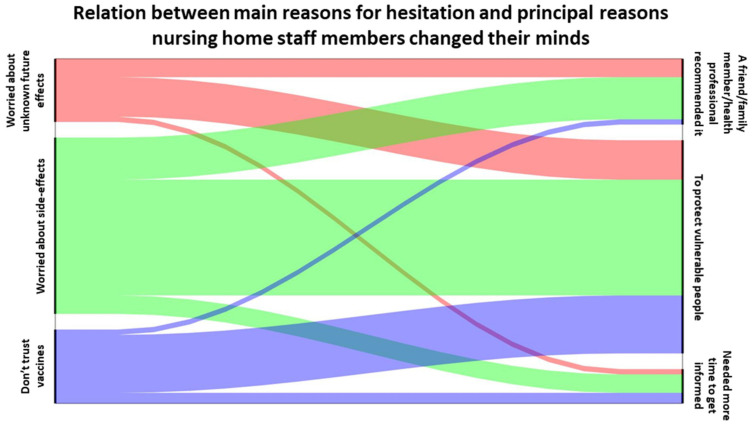
**Relation between the main reasons why nursing home staff members (NHS) hesitated to get vaccinated and the principal reasons they changed their minds**. Representation in the form of a Sankey diagram that link the three first reasons for hesitation (worried about side-effects, worried about unknown future effects and mistrust in vaccines) for vaccinated NHS that hesitated and NHS soon to be vaccinated (*N* = 119) with the three principal reasons that NHS changed their minds (it was recommended by a friend/family member/health professional, to protect vulnerable people, they needed more time to get informed).

**Table 1 vaccines-10-00598-t001:** Characteristics of 1142 Belgian nursing home staff members responding to the survey about the COVID-19 vaccine hesitancy during the fourth (between 26 July and 29 September 2021) and the fifth (between 29 September and 15 November 2021) visit.

	*N* (%)
**TOTAL**	1142
**Age**	
18–40	444 (38.9)
>40	629 (55.1)
**Gender**	
Male	158 (13.8)
Female	961 (84.2)
Unknown	23 (2)
**Region**	
Brussels	80 (7)
Flanders	678 (59.4)
Wallonia	384 (33.6)
**Type of job**	
Nursing	582 (51)
Paramedical	147 (12.9)
Cleaning staff	128 (11.2)
Catering	90 (7.9)
Administration	99 (8.7)
Hairdresser/pedicure	2 (0.2)
Other	47 (4.1)
**Comorbidity**	
Cardiovascular disease	35 (3.1)
Diabetes	35 (3.1)
Hypertension	120 (10.5)
Respiratory disorders	21 (1.8)
Immunosuppression	22 (1.9)
Cancer	7 (0.6)
None	893 (78.2)
**Influenza vaccination**	
Yes	546 (47.8)
No	549 (48.1)
**COVID-19 vaccine hesitancy**	
Vaccinated, no hesitation before	805 (70.5)
Vaccinated, hesitation before	276 (24.2)
Vaccinated soon, hesitation before	3 (0.2)
Not yet vaccinated, still hesitating	17 (1.5)
Refuse	41 (3.6)

Other type of job includes mainly logistic assistants (*N* = 20), animators (*N* = 10) and supervisors (*N* = 5). Due to missing values the numbers for some characteristics do not add up to the total number of the study population.

**Table 2 vaccines-10-00598-t002:** Association between vaccine hesitancy profiles and individual characteristics.

	Vaccinated, No Hesitation before	Vaccinated, Hesitation before	Not Yet Vaccinated or Soon, (Still) Hesitating	Refuse	*p*	*Vc*
	*N* (%)	*N* (%)	*N* (%)	*N* (%)		
**Age (years)**					<0.0001	0.14
18–40	281 (63.3)	137 (30.8)	9 (2)	17 (3.8)		
>40	480 (76.3)	121 (19.2)	9 (1.4)	19 (3)		
**Gender**					0.006	0.10
Female	661 (68.8)	249 (25.8)	17 (1.6)	34 (3.5)		
Male	129 (81.6)	21 (13.3)	2 (1.3)	6 (3.8)		
**Region**					<0.0001	0.18
Brussels	54 (67.5)	20 (25)	2 (2.5)	4 (5)		
Flanders	534 (78.8)	131 (19.3)	4 (0.6)	9 (1.3)		
Wallonia	217 (56.5)	125 (32.6)	14 (3.7)	28 (7.3)		
**Type of job**					0.0008	0.12
Medic. ^#^	495 (67.9)	199 (27.3)	14 (1.9)	21 (2.9)		
Non-medic. *	284 (77.6)	61 (16.7)	5 (1.4)	16 (4.4)		
**Comorbidity**					0.7233	0.03
No ^δ^	640 (71.7)	210 (23.5)	14 (1.5)	29 (3.2)		
Yes ^β^	139 (68.8)	50 (24.7)	5 (2.4)	8 (3.9)		
**Influenza vaccination**					<0.0001	0.33
Yes	466 (85.3)	79 (14.5)	1 (0.2)	0		
No	313 (57)	181 (33)	18 (3.2)	37 (6.7)		

The vaccine hesitancy profiles identified are: vaccinated nursing home staff members (NHS) that did not hesitate before vaccination; vaccinated NHS that hesitated before vaccination; unvaccinated NHS that will soon get vaccinated or are still hesitating to get vaccinated and unvaccinated NHS that have decided not to get vaccinated. Profiles are distributed by individual characteristics: age, gender, region, type of job with jobs divided into medical-related job (^#^ Medic.; nursing and paramedical) and non-medical-related job (* Non-medic.; cleaning staff, catering, administration, hairdresser/pedicure and other), Comorbidity (classified according to whether NHS self-reported zero (^δ^ No) or one or more (^β^ Yes) comorbidities) and influenza vaccination status in 2020. Numbers for some characteristics do not add up to the total number of the study population due to missing values. *p* value from Chi-squared test (χ^2^) and the measure of the degree of association with Cramer’s Phi (*Vc*).

**Table 3 vaccines-10-00598-t003:** Odds of vaccine hesitancy as a function of individual characteristics.

			Unadjusted	Adjusted
	No Hesitation before	Hesitation before/Still Hesitating/Refusing	OR [95% CI]	OR [95% CI]
	*N*	*N*		
**Age (years)**	761	312	0.96 [0.95, 0.97]	0.96 [0.94, 0.97]
**Gender**				
Female (ref)	661	300	1	1
Male	129	29	0.47 [0.31, 0.72]	0.46 [0.29, 0.75]
**Region**				
Brussels	54	26	1.75 [0.99, 3.10]	0.90 [0.44, 1.85]
Flanders (ref)	534	144	1	1
Wallonia	217	167	2.83 [2.07, 3.88]	2.22 [1.57, 3.12]
**Type of job**				
Medic. (ref)	495	234	1	1
Non-medic.	284	82	0.59 [0.44, 0.78]	0.68 [0.49, 0.95]
**Comorbidity**				
No (ref)	640	253	1	1
Yes	139	63	1.16 [0.84, 1.62]	1.82 [1.23, 2.69]
**Influenza vaccination**				
Yes (ref)	466	80	1	1
No	313	236	4.21 [3.12, 5.68]	3.86 [2.81, 5.30]

The vaccine hesitancy profiles are: vaccinated nursing home staff members (NHS) that did not hesitate before vaccination (No hesitation before = reference profile) and vaccinated NHS that hesitated before vaccination, unvaccinated NHS that will get vaccinated soon or not and hesitated or are still hesitating to get vaccinated and unvaccinated NHS that decided not to get vaccinated (Hesitation before/still hesitating/refusing). Profiles are distributed by individual characteristics: age, gender, region, type of job with jobs divided into medical-related job (Medic.; nursing and paramedical) and non-medical-related job (Non-medic.; cleaning staff, catering, administration, hairdresser/pedicure and other), Comorbidity (classified according to whether NHS self-reported zero (No) or one and more (Yes) comorbidities) and influenza vaccination status in 2020. Data are shown as unadjusted odds ratio (OR) with 95% confidence interval (95%CI) and adjusted OR for all covariates (multivariate analysis) with 95%CI. The odds ratios are estimated based on a GEE analysis with exchangeable covariance structure, taking the clustered nature of the staff within NH into account.

## Data Availability

Data is available on request.

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
