# Peer review of "Vaccine Hesitancy towards the COVID-19 Vaccine in a Random National Sample of Belgian Nursing Home Staff Members"

_vaccines, 2022, doi:10.3390/vaccines10040598_

Round 1

Reviewer 1 Report

Materials and Methods

  1. The Authors should clarify how has been determined the sample size.
  2. The Authors should clarify about the face-validity testing of the questions with an explanation of the validity of the content of the questions with regard to the research aims. The Authors should clarify how they had estimated the reliability, or internal consistency, of the questions by using, for example the Cronbach’s alpha in order to measure whether or not a score is reliable.
  3. It is not given any information whether the participant was informed about the use and anonymization of the data and that survey responses guarantee the anonymity of each participant.
  4. It is not given any information whether participants were not able to continue to the next question of the questionnaire if they failed to provide a response to an item.
  5. It should be clarified whether the participants have received any gift or monetarily compensated.
  6. It is not given any information regarding a pilot study for testing the survey questionnaire.
  7. The statistical analysis is not, strictly speaking, adequate. The absence of a multivariate statistical analysis is the major concern, because it would be particularly relevant to measure the associations between several characteristics and the outcomes of interest. Moreover, it should be indicated if the tests were one-side or two-sided.

Results

  1. No information is given about the subjects who did not agree to participate. Was there any attempt to quantify the response bias: information about non-responders. It would be useful to have some kind of indication of comparability with non-respondents. Is there any population-based data available? How did they differ from those in the sample, how representative is the sample and were the findings representative of the population?

Discussion

  1. The pivotal role of healthcare providers and scientific journals as sources of information with a positive impact towards vaccination attitudes should be stressed and studies supporting this statement should be added. For example, the following articles should be cited Di Giuseppe et al. Expert Rev Vaccines 2021;20:881-9; Wang et al. Vaccines (Basel) 2021;9(3):29.
  2. It has been observed that many non-vaccinated respondents have serious concerns about the fear of unknown future effects and comparison with similar previous studies should be added.
  3. There is a lack of comparison with the results of recent studies conducted in other geographic areas. The work should therefore be enriched in such a way as to become self-supporting by photographing the context and what is around it.
  4. The paragraph regarding the limitations of the study should discuss all limits such as, for example, the study design, the recall bias, and the social desirability bias.

Tables

  1. In the Tables the Unknown should be removed and should be clarified that the number of respondents is not the same for all variables. Add a footnote Numbers for some characteristics do not add up to the total number of the study population due to missing values.
  2. Table 2 and Figures 1 and 2 should be deleted.

References

  1. a) The manuscript is still not well referenced. The References list is not updated, since several articles conducted in different countries and published on peer-reviewed journals have been not included.

Reviewer 2 Report

The manuscript is a prospective cohort study that quantifies the COVID-19 vaccine hesitancy rates and analyses its determinants among Belgian nursing home staff members. Data is interesting, but in its current form is hard to read. Some comments to improve the manuscript.

Methods.

You should better describe how the four groups you analyzed were coded (i.e., 1. Vaccinated, no hesitation before; 2. Vaccinated, hesitation before; 3. Not yet vaccinated or soon, (still) hesitating; 4. Refuse) as it is very hard to read. Probably supplementary figure 1 should be included in the text.

How were variables selected to be part of the final model?

In addition, collapsing hesitation before/still hesitating/refusing into one category needs to be conceptually justified.

Table 2. This table is not clear enough.

Why “get vaccinated soon but hesitated” is under “Total unvaccinated”?

Figure 1. Probably a histogram with horizontal stacked bars would be more intuitive and easier to interpret.

Table 4. The comparisons are unclear. What does each OR refer to?

Please separate the legend of table 2 and the main text (page 6).

Discussion.

This section needs to be re-organized. For example, strengths and limitations should go at the end of the manuscript.

Please do not repeat results in this section (line 256-251) but focus on how your findings align or not with the current literature and what is the overall meaning of your study.

References.

Bibliography is a bit poor. Vaccine hesitancy among healthcare workers has been largely described in Europe.

Reviewer 3 Report

the manuscript aims to explore reasons for covid-19 vaccine hesitancy among Belgian nursing home staff.
It is a relevant and original topic since COVID-19 is still an important health challenge
It is the first study among Belgian nursing home staff study, extensively exploring reasons of hesitancy.
The method is appropriate and clearly reported. Please add description of hesitancy and refusal, as well the definition of variables used in the analysis. Lastly, please specify the variables used in the adjusted model. Add this info both in the statistical analysis section and as a caption of the table.
The conclusions are consistent with the evidence and arguments presented and do they address the main question posed

The references are appropriate

Table 1 and 2 may be condensed in one

Round 2

Reviewer 1 Report

The Authors have fully addressed the concerns that have been raised.

Author Response

Dear Reviewer #1,

We would like to thank you for the thorough review of the manuscript newly entitled “Vaccine hesitancy towards the COVID-19 vaccine in a random national sample of Belgian nursing home staff members” submitted to Vaccines. We feel that the proposed modifications made during the first review have helped us to improve the paper.

After first reviewing, it was stated that we have fully addressed the concerns that have been raised. 

Yours sincerely,

Béatrice Scholtes

Reviewer 2 Report

All comments have been addressed.

Author Response

Dear Reviewer #2,

We would like to thank you for the thorough review of the manuscript newly entitled “Vaccine hesitancy towards the COVID-19 vaccine in a random national sample of Belgian nursing home staff members” submitted to Vaccines. We feel that the proposed modifications made during the first review have helped us to improve the paper.

After first reviewing, it was stated that we have fully addressed the concerns that have been raised. 

Yours sincerely,

Béatrice Scholtes